# A Block-Based Adaptive Decoupling Framework for Graph Neural Networks

**DOI:** 10.3390/e24091190

**Published:** 2022-08-25

**Authors:** Xu Shen, Yuyang Zhang, Yu Xie, Ka-Chun Wong, Chengbin Peng

**Affiliations:** 1College of Information Science and Engineering, Ningbo University, Ningbo 315211, China; 2Department of Computer Science, City University of Hong Kong, Hong Kong 999077, China

**Keywords:** graph neural networks, block-based methods, network decoupling, adaptive receptive fields

## Abstract

Graph neural networks (GNNs) with feature propagation have demonstrated their power in handling unstructured data. However, feature propagation is also a smooth process that tends to make all node representations similar as the number of propagation increases. To address this problem, we propose a novel Block-Based Adaptive Decoupling (BBAD) Framework to produce effective deep GNNs by utilizing backbone networks. In this framework, each block contains a shallow GNN with feature propagation and transformation decoupled. We also introduce layer regularizations and flexible receptive fields to automatically adjust the propagation depth and to provide different aggregation hops for each node, respectively. We prove that the traditional coupled GNNs are more likely to suffer from over-smoothing when they become deep. We also demonstrate the diversity of outputs from different blocks of our framework. In the experiments, we conduct semi-supervised and fully supervised node classifications on benchmark datasets, and the results verify that our method can not only improve the performance of various backbone networks, but also is superior to existing deep graph neural networks with less parameters.

## 1. Introduction

Graph-structured data are widely used in various fields, such as social networks [1,2], knowledge graphs [3,4], and citation networks [5,6]. Graph Neural Networks (GNNs) have been widely used and have achieved state-of-the-art performance in many related applications, such as node classification [5,6,7,8], link prediction [9,10,11], and graph classification [12,13]. Feature propagation is a simple, efficient, and powerful GNN paradigm [14,15]. The main idea behind it is to obtain new node representations by stacking multiple GNN layers to aggregate the neighbor information of nodes using nonlinear transformations [16]. Graph Convolutional Network [5] is one of the representative methods, which iteratively aggregates the features of neighboring nodes using a normalized adjacency matrix. However, it can only work with two to four layers, and when the model is deeper, the representation ability will degrade rapidly. The reason is that when the GCN layers are continuously stacked, the representations of nodes eventually converge to a specific value and become indistinguishable [17]. Some studies believe that GCN is a smoothing operation on the graph using the Laplacian operator [18], so the above problem is also called the over-smoothing problem [19].

In order to learn high-level node representations in large, sparsely connected graphs, we have to increase model depth. For this sake, approaches that can alleviate over-smoothing have been developed.

Some approaches simply modify the connections between GNN layers, such as using residual connections and identity maps [20,21], skip connections [17], inception structures [22], and self-attention for different neighbors [23,24]. However, the increase in performance is still limited [21].

In addition to these approaches, network decoupling is an important way to alleviate over-smoothing. Traditional GNNs map from input to output space using feature transformation operation after feature propagation. However, recent studies have shown that too many feature transformations can increase unnecessary redundant computation [25], cause over-fitting [22], and accelerate over-smoothing [18,26]. Decoupled GNNs can solve such problems by separating the transformation and propagation process, such as propagating features multiple times and then performing a few feature transformations [26,27,28,29], or reversely [30,31]. Since the feature propagation process does not involve parameter training, decoupled GNNs are also beneficial to the offline computation of the feature propagation process for some giant graphs, which significantly reduces the training time.

However, these approaches can not adaptively learn the number of optimal transformations.

Another way to solve over-smoothing is to use a flexible receptive field for each node. Traditional GNNs usually use a fixed receptive field, and the node representations output by the last layer of a model only consider the neighborhood within a specific distance. Thus, information during the propagation process is not fully utilized and not adjustable [32]. Some works try to make the receptive field of the node adaptively adjusted by combining the outputs from different GNN layers [27,28,33]. Some methods choose to concatenate multiple levels of features together [34], and some methods choose to add these features by weights [28,30]. However, these approaches bring additional computational complexity.

In this work, we propose a novel decoupling approach, called a Block-Based Adaptive Decoupling (BBAD) Framework, to improve the performance further with less computational complexity for deep networks. We use decoupled blocks to replace multi-layer GNNs for feature propagation in this framework. Different backbone networks can be used in each block, and we use an attention mechanism to assign weights to adjust the receptive field. We also propose a method to automatically adjust the number of layers in each block based on identity mapping and L1 regularization so that it can adaptively balance the number of operations for feature transformation and propagation. Experiments for semi-supervised and fully supervised node classification show that our framework can improve the performance of backbone networks significantly and outperform existing deep models with fewer parameters. The main contributions of this paper are as follows:We propose an adaptive block-based decoupling framework. It can combine shallow models into a deep one, producing high-level feature representations and providing flexible receptive fields for different nodes while reducing over-smoothing and over-fitting. We also propose a layer regularization approach to automatically adjust the propagation depth in decoupling blocks to control the decoupling rate.We prove that the traditional coupled GNNs are more likely to suffer from over-smoothing when they become deep. We explore the importance of an appropriate decoupling rate and demonstrate the diversity of outputs from different blocks of our framework.We conduct semi-supervised and fully supervised node classifications on benchmark datasets. The results verify that our method can not only improve the ability of various backbone networks to acquire deep features, but also outperform existing deep graph neural networks with fewer parameters.

## 2. Related Work

GNNs typically aim to find a convolution kernel suitable for graph structure data. Some researchers have proven that the convolution operation on a graph could be approximated by the k-order polynomial of the Laplace operator of the graph [5,35]. For example, Kipf et al. proposed that the graph convolution network (vanilla GCN) simplifies the previous graph convolution model by the first-order approximation of the k-order polynomial [5], and the representation of the graph convolution layer is obtained as
(1)$$H^{l+1}=\sigma \left(\tilde{P}H^{l}W^{\left(l\right)}\right),$$
where $\tilde{A}=A+I$, $\tilde{P}=D^{-\frac{1}{2}}\tilde{A}D^{-\frac{1}{2}}$ is a normalized adjacency matrix, $H^{l+1}$ is the feature matrix of layer *l*, *W* represents the learnable parameters of the linear transformation layer, and σ represents a nonlinear activation function, such as RELU. GCN aggregates neighboring node features by iteratively stacking multiple graph convolutional layers. GCN makes the convolution operation on graphs simple, but as mentioned above, GCN suffers from over-smoothing, so that GCN cannot take advantage of deep neural networks to learn high-level representations.

Many approaches have been proposed to solve this issue. Modifying the structure of feature propagation in the model is one of them, as shown in Figure 1. For example, JKnet analyzed the failures in GCN from the spatial domain and proposed a feasible deep GNN model, which adopts the structure of dense connections for feature propagation [17]. It concatenates the outputs from all the layers together, $H^{l}=\left[H^{l-1},\cdots H^{0}\right]$, and solves the over-smoothing problem by combining node representations with different hops.

Some research demonstrates the necessity and superiority of decoupled GNNs theoretically, and removes feature transformation operations while only retaining feature propagation layers, and the final classification layer [25]. Decoupling GNNs can improve the flexibility of feature propagation and remove redundant parameters, which helps to improve the ability of GNNs to acquire deep features. The feature propagation layer can be expressed as
(2)$$H^{\left(l\right)}=SH^{\left(l-1\right)},$$
where $\tilde{A}=A+I$, $\tilde{S}=D^{-\frac{1}{2}}\tilde{A}D^{-\frac{1}{2}}$, $H^{\left(l\right)}$ is the graph convolution output feature of the *l*th layer.

On the other hand, Klicpera et al. argue that the size of the aggregated neighborhood required in GNNs and the depth of the feature transformation are two completely orthogonal aspects, so they propose APPNP based on personalized PageRank to solve the problem of over-smoothing [31]. Formally, the definition of the aggregation layer is as follows
(3)$$H^{\left(0\right)}=\sigma \left(WX\right),$$
(4)$$H^{\left(l+1\right)}=\alpha \tilde{P}H^{\left(l\right)}+\left(1-\alpha \right)H^{\left(0\right)},$$
where $\tilde{P}$ is the same normalized adjacency matrix as in GCN, $\alpha \in \left[0,1\right]$, and *W* represents the learnable parameter and is shared for each APPNP layer to decouple the model. Thus, multi-layer information aggregation performed from multi-hop neighbors will not significantly increase the computational cost. To avoid over-smoothing, the input feature is partially maintained by adding skip connections between the input layer and the current layer.

DAGNN adopts a similar shared feature transformation method [30]. It performs feature transformation on the initial features of nodes, and then performs feature propagation. The outputs of the different layers are adaptively fused as follows
(5)$$H_{final}=\sum_{l=0}^{K}\theta_{l}H^{\left(l\right)},$$
where $H^{\left(l\right)}$ represents the node representation of the output of the *l*th layer. By fusing the node features of different neighborhoods, DAGNN effectively alleviates the over-smoothing problem at the cost of computational complexity.

## 3. Block-Based Adaptive Decoupling Framework

We introduce our proposed adaptive block-based decoupling framework in this section. Using blocks as the basic feature propagation units enables our architecture to be flexible and versatile enough to be applied to different backbone GNNs.

### 3.1. Main Model

Our proposed framework comprises three parts: feature transformation, feature propagation with adaptive depth, and flexible node receptive fields. An illustration of our proposed framework is provided in Figure 2.

#### 3.1.1. Initial Feature Transformation

Initial feature transformation is performed on the input features of nodes through a single layer. As shown in Equation (Equation 6),
(6)$$Z=\sigma \left(WX\right),$$
where *W* is the linear transformation parameter shared by all feature propagation blocks, *X* is the input feature of nodes, and σ is the activation function. We use RELU by default. This step is similar to other decoupled GNN models.

#### 3.1.2. Feature Propagation with Adaptive Depth

The core of GNN is feature propagation, because feature transformation alone cannot use the neighborhood information. In order to alleviate over-smoothing and over-fitting due to too many feature transformations in each block, we remove all feature transformation operations and only retain the feature propagation operations between neighboring nodes. Therefore, the *k*-th feature propagation layer in each block can be written as follows
(7)$$m_{v}^{\left(k\right)}=f_{M}^{\left(k\right)}\left\{h_{u}^{k-1},u\in N\left(v\right)\right\},$$
(8)$$h_{v}^{\left(k\right)}=f_{C}^{\left(k\right)}\left(m_{v}^{k},h_{v}^{k-1}\right),$$
where $h_{v}^{\left(k\right)}$ is the feature representation of the local node *v* after *k* feature propagation operations, $f_{M}$ is a neighborhood information aggregation function, and $m_{u}^{\left(k\right)}$ is the feature representation of node *u*; $N\left(v\right)$ contains the neighboring nodes of *v*, and $f_{C}$ is a function that decides how to combine $h_{v}^{\left(k-1\right)}$ and $m_{v}^{\left(k\right)}$. Different GNNs have different definitions of $f_{M}$ and $f_{C}$; for example, in GCN, $m_{v}^{\left(k\right)}=SUM\left\{h_{u}^{k-1},u\in N\left(v\right)\right\}$ and $h_{v}^{\left(k\right)}=ADD\left(m_{u}^{k},h_{v}^{k-1}\right)$.

The depth of feature propagation significantly affects the performance. Thus, each block should control its depth to achieve the best performance. For this sake, we adopt an identity map to control the depth of feature propagation of a single block. The overall node representation in a block after *k* layers can be written as
(9)$${\hat{H}}^{\left(k\right)}=\beta^{\left(k\right)}H^{\left(k\right)}+\left(1-\beta^{\left(k\right)}\right){\hat{H}}^{\left(k-1\right)},$$
where $H^{\left(k\right)}$ is the feature matrix composed of all node representation after feature propagation for *k* times; that is, $H^{\left(k\right)}=\left[h_{v_{1}}^{\left(k\right)},h_{v_{2}}^{\left(k\right)},\cdots \right]$. ${\hat{H}}^{\left(k-1\right)}$ is the output from the previous layer, and ${\hat{H}}^{\left(k\right)}$ is the final output of the *k*-th layer. $\beta^{\left(k\right)}$ corresponds to the control parameters used by the *k*-th layer for identity mapping. When $\beta^{\left(k\right)}$ is close to zero, it means that the operation of the *k*-th layer will be skipped, and the input is directly mapped to the output. When $\beta^{\left(k\right)}$ is close to one, it means that the operation of the *k*-th layer will be passed to the next layer. In this manner, the depth of the entire block can be adaptively tuned by changing the value of β. We also add an L1-regularization term $\sum_{k}^{}\left|\beta_{k}\right|$ to control the sparseness of β. Continuously minimizing the loss function through backpropagation can adaptively optimize depth.

A decoupling block comprises multiple feature propagation layers without any feature transformation layer. The feature propagation within the block is carried out layer by layer, and any propagation structure as shown in Figure 1 can be used between layers to improve the propagation ability. The calculation is as follows:(10)$$B^{\left(i\right)}=prop\left(B^{\left(i-1\right)}\right),$$
where prop represents multi-layer feature propagation, and $B^{\left(i\right)}$ represents the output of the *i*-th block and also the input of the next block.

We also need to choose an appropriate ratio of the feature propagation layer number to the feature transformation layers number; namely, the decoupling rate. In our framework, we fix the number of feature transformation layers to be one, which means that only a single feature transformation is performed at the end of each block. Since the number of feature propagation of each block is adaptively changed, the decoupling rate of each block can be adjusted automatically. These two steps can be written as follows:(11)$${\hat{B}}^{\left(i\right)}=\sigma \left(W^{\left(i\right)}B^{\left(i\right)}\right),$$
where $W^{\left(i\right)}$ represents the linear transformation parameters corresponding to the *i*-th block, and ${\hat{B}}^{\left(i\right)}$ represents the feature representations that will be taken as input to the adaptive node receptive fields and not passed to the next block.

#### 3.1.3. Adaptive Node Receptive Fields

The output of each block in the framework corresponds to the node representation after feature propagation with different hops. In the adaptive adjustment of node receptive field, we aim to assign receptive fields of different sizes to different nodes, which can be achieved by aggregating features from low-order and high-order neighbors from different blocks using different weights. From the perspective of spectral domain analysis, it works similarly to a filter for different frequencies to better use the signals on the graph at various frequencies. The calculation of this step is as follows
(12)$$H_{final}=\sum_{l=0}^{K}\alpha_{l}{\hat{B}}^{\left(l\right)},$$
where ${\hat{B}}^{\left(l\right)}$ is the final output of the *l*-th block, $\alpha_{l}$ measures the impact of the output of the *l*-th block on the final node representation, and $H_{final}$ is the final output of the entire framework.

To determine weights, we use a recursive attention mechanism. It recursively calculates how much discriminative information the current feature can bring to the previous combined features to guide the weight assignment. Its calculation form is as follows
(13)$${\tilde{B}}^{\left(l\right)}={\hat{B}}^{\left(l\right)}\|\sum_{k=0}^{l-1}\alpha^{\left(k\right)}{\hat{B}}^{\left(k\right)},$$
(14)$${\tilde{\alpha }}^{\left(l\right)}=\delta \left({\tilde{B}}^{\left(l\right)}\cdot s\right),$$
(15)$$\alpha^{\left(l\right)}=e^{{\tilde{\alpha }}^{\left(l\right)}}/\sum_{k=0}^{K}e^{{\tilde{\alpha }}^{\left(k\right)}},$$
where ‖ means the concatenation of two block outputs, and *s* is a learnable vector. ${\tilde{B}}^{\left(l\right)}$ combines features from different propagation hops. If it contains most of the information in $\sum_{k=0}^{l-1}\alpha^{\left(k\right)}{\hat{B}}^{\left(k\right)}$; this means that the features of the neighborhood are very smooth, and ${\tilde{B}}^{\left(l\right)}$ should be assigned with a smaller weight to avoid over-smoothing. On the other hand, if the weight assigned to ${\tilde{B}}^{\left(l\right)}$ is large, it means that ${\tilde{B}}^{\left(l\right)}$ can contribute for more discriminative information.

Our proposed framework can be applied to multiple graph-related downstream tasks, and eventually, we will update all parameters in the whole architecture by optimizing the loss function. Taking node classification as an example, we use the cross-entropy to measure the differences between the softmax predictions and the ground-truth labels. The final loss function can be as follows
(16)$$L=-\sum_{i\in V_{i}}^{}\sum_{j}^{}Y_{ij}log\left(softmax{\left(H_{final}\right)}_{ij}\right)+\lambda \sum_{i}^{}\left|\beta_{i}\right|,$$
where the first term is the cross-entropy loss function, and λ is to balance the two loss terms.

### 3.2. Comparing with Existing Decoupling GNNs

In this section, we compare the similarities and differences between our framework and existing decoupling approaches.

**Comparison with Deep Adaptive Graph Neural Network** (**DAGNN**) [30]: The decoupling method of this approach is to transform features first and then propagate these features. Our framework can adopt different feature propagation structures in each block, as shown in Figure 1, while DAGNN can only perform simple layer-by-layer propagation. DAGNN is forced to output the results of each feature propagation layer. Although it can fuse multi-hop neighborhoods, it ignores gains from the layer-to-layer connections, such as skip residual connections, initial residual connections, etc. Meanwhile, our model is block-based, utilizing the output of each block to construct an adaptive receptive field, and different propagation structures can be flexibly adopted within the block. So, our framework is more lightweight when computing adaptive receptive fields. For example, if 64 layers of feature propagation are to be performed, DAGNN needs to assign weights for 64 weights. In comparison, our model propagates eight layers per block and finally, it only needs to assign weights for eight features. Experimental results also show that our method outperforms DAGNN in both performance and complexity.**Comparison with Decoupled GCN** (**DGCN**) [26]: Alternatively, DGCN propagates features first and then transforms them. The number of feature propagation layers is the same as that of the feature transformation layer. DGCN assigns a parameter to each layer to control the proportion of feature transformation, so that the parameters of the model are generally unchanged. Our framework uses adaptive decoupling blocks to reduce the number of model parameters.

### 3.3. Theoretical Analysis

We can consider the adaptive node receptive field as an ensemble approach by treating different GNN network layers as different basic learners. Since the ensemble effectiveness of basic learners depends on the diversity of learned features, we demonstrate that our approach can provide diversified basic solutions before feeding into Equation (Equation 12). In this analysis, we assume that node features are generated using normal distributions with varying parameters, and the probability that a node shares the same feature distribution parameters with another node is inversely proportional to their distance. Hereby, we define the probability that two nodes have the same distribution as $\gamma^{k}$, where γ∈[0,1], and *k* is the distance between two nodes.

Without loss of generality, we can define two independent distributions as follows, for a given node with its feature generated from $N\left(\mu_{A},\sigma_{A}\right)$. Thus, the proportion of its *k*-hop neighbors obtaining features from the same distribution is $\gamma^{k}$, and the features of the remaining *k*-hop nodes are generated from $N\left(\mu_{B},\sigma_{B}\right)$.

**Theorem** **1.**
*The correlation between the output of the $k_{1}$-th feature propagation block and that of the $k_{2}$-th block is*

(17)
$$\begin{array}{c} \frac{\mu_{k_{1}}\left(1-\mu_{k_{2}}\right)\sigma_{k_{2}}^{2}+\mu_{k_{2}}\left(1-\mu_{k_{1}}\right)\sigma_{k_{1}}^{2}}{\left(\sigma_{k_{1}}^{2}+\sigma_{k_{2}}^{2}\right)\sigma_{k_{1}}\sigma_{k_{2}}}, \end{array}$$

*where*

(18)
$$\begin{array}{cc} \mu_{k} & =\gamma^{k}\mu_{A}+{\left(1-\gamma \right)}^{k}\mu_{B}, \end{array}$$


(19)
$$\begin{array}{cc} \sigma_{k} & =\gamma^{2k}\sigma_{A}^{2}+{\left(1-\gamma \right)}^{2k}\sigma_{B}^{2}, \end{array}$$


*with $k=k_{1}$ or $k_{2}$.*


**Proof.** We use $X_{A}^{i}∼N\left(\mu_{A},\sigma_{A}\right)$ and $X_{B}^{i}∼N\left(\mu_{B},\sigma_{B}\right)$ to represent the corresponding random variables, and *i* is an identifier. For the *k*-th block, we define the number of aggregated features from neighborhoods as $d_{k}$. According to the definition of feature propagation, the aggregated feature can be represented as follows
(20)$$\begin{array}{cc} X_{k} & =\frac{1}{d_{k}}\left(\sum_{i=0}^{\gamma^{k}d_{k}}X_{A}^{i}+\sum_{i=0}^{{\left(1-\gamma \right)}^{k}d_{k}}X_{B}^{i}\right). \end{array}$$
The corresponding expectation and variation are
(21)$$\begin{array}{cc} E[X_{k}] & =\gamma^{k}\mu_{A}+{\left(1-\gamma \right)}^{k}\mu_{B}, \end{array}$$
(22)$$\begin{array}{cc} D[X_{k}] & =\gamma^{2k}\sigma_{A}^{2}+{\left(1-\gamma \right)}^{2k}\sigma_{B}^{2}. \end{array}$$Therefore, the covariance of the output from the $k_{1}$-th block and that from the $k_{2}$-th block is
(23)$$\begin{array}{cc} \; & Cov\left(X_{k_{1}},X_{k_{2}}\right) \end{array}$$
(24)$$\begin{array}{cc} = & E[\left(X_{k_{1}}-E\left[X_{k_{1}}\right]\right)\left(X_{k_{2}}-E\left[X_{k_{2}}\right]\right)]\\ = & E\left[X_{k_{1}}X_{k_{2}}\right]-E\left[X_{k_{1}}\right]\mu_{X_{k_{2}}}-E\left[X_{k_{2}}\right]\mu_{X_{k_{1}}} \end{array}$$
(25)$$\begin{array}{cc} \; & +\mu_{X_{k_{1}}}\mu_{X_{k_{2}}} \end{array}$$
(26)$$\begin{array}{cc} = & E\left[X_{k_{1}}X_{k_{2}}\right]-\mu_{X_{k_{1}}}\mu_{X_{k_{2}}} \end{array}$$
(27)$$\begin{array}{cc} = & \frac{\mu_{k_{1}}\sigma_{k_{2}}^{2}+\mu_{k_{2}}\sigma_{k_{1}}^{2}}{\sigma_{k_{1}}^{2}+\sigma_{k_{2}}^{2}}-\mu_{X_{k_{1}}}\mu_{X_{k_{2}}} \end{array}$$
(28)$$\begin{array}{cc} = & \frac{\mu_{k_{1}}\left(1-\mu_{k_{2}}\right)\sigma_{k_{2}}^{2}+\mu_{k_{2}}\left(1-\mu_{k_{1}}\right)\sigma_{k_{1}}^{2}}{\sigma_{k_{1}}^{2}+\sigma_{k_{2}}^{2}}. \end{array}$$Thus, the correlation between the outputs of two blocks is
(29)$$\begin{array}{cc} \rho_{X_{k_{1}}X_{k_{2}}}= & \frac{Cov\left(X_{k_{1}},X_{k_{2}}\right)}{\sqrt{D\left(X_{k_{1}}\right)}\sqrt{D\left(X_{k_{2}}\right)}} \end{array}$$
(30)$$\begin{array}{cc} = & \frac{\mu_{k_{1}}\left(1-\mu_{k_{2}}\right)\sigma_{k_{2}}^{2}+\mu_{k_{2}}\left(1-\mu_{k_{1}}\right)\sigma_{k_{1}}^{2}}{\left(\sigma_{k_{1}}^{2}+\sigma_{k_{2}}^{2}\right)\sigma_{k_{1}}\sigma_{k_{2}}}. \end{array}$$
where $\mu_{k_{1}}$, $\mu_{k_{2}}$, $\sigma_{k_{1}}$, and $\sigma_{k_{2}}$ are calculated by Equations (Equation 18) and (), with $k=k_{1}$ or $k_{2}$.Theorem 1 is proved. □

Next, we demonstrate the relationship between the decoupling rate and the over-smoothing phenomenon. As matrix $\tilde{A}$ is symmetric, its eigenvalues $\lambda_{1}\leq \cdots \leq \lambda_{N}$ are all real numbers.

**Lemma**  **1.**
*(Augmented Spectral Property [18]) When the multiplicity of the largest eigenvalue $\lambda_{N}$ is M, there are the following properties: $-1<\lambda_{1}$, $\lambda_{N-M}<1$, and $\lambda_{N-M+1}=\cdots =\lambda_{N}=1$.*


**Definition** **1.**
*(M-dimensional sub-space [36]) An M-dimensional$\left(M<N\right)$ sub-space in $R^{N\times C}$ is defined as follows:*

(31)
$$M:=\left\{H\in R^{N\times C}|H=\tilde{E}C,C\in R^{M\times C}\right\},$$


*where $\tilde{E}=\left\{{\tilde{e}}_{1},\cdots ,{\tilde{e}}_{M}\right\}\in R^{N\times M}$ contains the bases of the largest eigenvalue of $\tilde{A}$ in Lemma 1. Namely, ${\tilde{e}}_{m}={\tilde{D}}^{\frac{1}{2}}u_{m}$, where $u_{m}\left(i\right)=1$ if node i belongs to the m-th connected components, and vice versa. The M subspace only contains the degree information of the nodes, and is the space to which the node representation converges when oversmoothing occurs.*


**Lemma**  **2.**
*(Distance measure [18]) The distance between matrix $H\in R^{N\times M}$ and M is $d_{M}\left(H\right):=\inf_{Y\in M}{∥H-Y∥}_{F}$, with the following properties*

(32)
$$d_{M}\left(\tilde{A}H\right)\leq \eta d_{M}\left(H\right),$$


(33)
$$d_{M}\left(HW\right)\leq \varphi d_{M}\left(H\right),$$

*where η is the second largest eigenvalue of $\tilde{A}$, φ is the supremum of all singular values of all $W_{l}$, and both of them are less than one.*


We define that the output of a vanilla GCN with *l* feature propagation layers is $H_{l}$. We also define the output of a block-based framework with multiple vanilla GCNs as $B_{k}$, with decoupling rate $R_{d}=1-\frac{k}{l}$.

**Theorem** **2.**
*With the definitions above, the following properties hold*

*$d_{M}\left(H_{l}\right)\leq {\left(\eta \varphi \right)}^{l}d_{M}\left(X\right)$;*

*$d_{M}\left(B_{k}\right)\leq \eta^{l}\varphi^{\left(1-R_{d}\right)l}d_{M}\left(X\right)$.*



**Proof.** For layer-based GCN, using Equations (Equation 32) and (Equation 33), we can have
(34)$$\begin{array}{cc} d_{M}\left(H_{l+1}\right) & \leq d_{M}\left(\tilde{A}H_{l}W_{l}\right) \end{array}$$
(35)$$\begin{array}{cc} \; & \leq \eta d_{M}\left(H_{l}W_{l}\right) \end{array}$$
(36)$$\begin{array}{cc} \; & \leq \eta \varphi d_{M}\left(H_{l}\right). \end{array}$$
Then, we have the relationship between the input feature *X* and $H_{l}$
(37)$$d_{M}\left(H_{l}\right)\leq {\left(\eta \varphi \right)}^{l}d_{M}\left(X\right).$$For the block-based decoupling GCNs, the relationship between the blocks is
(38)$$\begin{array}{cc} d_{M}\left(B_{i+1}\right) & \leq d_{M}\left({\tilde{A}}^{\left(l/k\right)}B_{i}W_{i}\right)\\ \; & \leq \eta^{\left(l/k\right)}\varphi d_{M}\left(B_{i}\right), \end{array}$$
so the relation between $B_{k}$ and the input feature *X* is
(39)$$\begin{array}{cc} d_{M}\left(B_{k}\right) & \leq {\left(\eta^{\left(l/k\right)}\varphi \right)}^{k}d_{M}\left(X\right) \end{array}$$
(40)$$\begin{array}{cc} \; & \leq \eta^{l}\varphi^{k}d_{M}\left(X\right), \end{array}$$
(41)$$\begin{array}{cc} d_{M}\left(B_{k}\right) & \leq \eta^{l}\varphi^{\left(1-R_{d}\right)l}d_{M}\left(X\right) \end{array}$$Theorem 2 is proven. □

From this theorem, we can find that, given the same number of feature propagation layers, our approach is less likely to converge to the over-smoothing state.

For different GNNs, $d_{M}\left(H_{l}\right)$ is different, and so is $d_{M}\left(B_{k}\right)$. For example, for ResGCN, $H_{l}$ is calculated as follows
(42)$$H^{\left(l\right)}=\alpha H^{\left(l-1\right)}+W^{\left(l-1\right)}\tilde{A}H^{\left(l-1\right)},$$
Therefore, we have
(43)$$d_{M}\left(H_{l}\right)\leq {\left(\varphi \eta +\alpha \right)}^{l}d_{M}\left(X\right),$$
(44)$$d_{M}\left(B_{k}\right)\leq \left({\left(\eta +\alpha \right)}^{l}+\varphi^{\left(1-R_{d}\right)l}\right)d_{M}\left(X\right).$$

The calculation method is the same as that of Equations (Equation 34) and (Equation 37)–(Equation 39), so for GNNs with different propagation structures, the required decoupling rates are different. Our architecture solves this problem by adaptively adjusting the decoupling rate.

### 3.4. Importance of the Decoupling Rate

In this section, we discuss the importance of the decoupling rate with respect to the number of layers in different models. In traditional GNNs, each feature propagation layer is followed by a feature transformation layer, and the two operations are fully coupled. When we allow multiple feature propagation or transformation layers, we can define the decoupling rate as follows:(45)$$R_{d}=1-\frac{L_{t}}{L_{p}},$$
where $L_{p}$ is the number of feature propagation layers and $L_{t}$ is the number of feature transformation layers. Typically, as non-decoupled GNNs force feature propagation and transformation to be performed simultaneously, we can have $L_{p}=L_{t}$. Using decoupled GNNs usually increases the number of propagations to expand the range of the aggregated neighborhood, so that $L_{p}>L_{t}$. Therefore, with $L_{p}>=L_{t}$, the decoupling rate is in $\left[0,1\right]$. Thus, $R_{d}$ is one minus the ratio of feature transformation layers to the feature propagation layers.

Graphs are a kind of complex data, and feature propagation can effectively utilize the topological characteristics of the node’s local neighborhoods, while feature transformation can utilize the correlation between the node labels and features. Some studies believe that too many feature transformations can damage the performance of GNNs [25,31], so only a small number of feature transformations are retained after decoupling. For example, SGC removes all the feature transformations before the output layer in GCN [25]. However, it can achieve good performance with shallow layers, but does not perform well when using deep layers. In the following examples, we demonstrate that an appropriate decoupling rate is essential when the number of network layers varies.

We conduct node classification experiments on Cora and Citeseer, which are graphs with sparse edges and which are likely to suffer from over-smoothing. In Figure 3a,b, we can find that when the number of layers indicated by the horizontal axis changes, the best choice of decoupling rate indicated by the vertical axis also changes. In Figure 3c, when the depth is 4, 9, or 10, ResGCN requires one 100% decoupling rate, which means that removing all feature transformation layers before the output layer is best. The decoupling rate required by JKnet is less than that by ResGCN. The relationship between the two is complex and cannot be described by a simple expression.

We can also interpret these figures from another perspective. When we use a total of 10 layers, we can see from Figure 3c that the accuracy varies as the number of feature transformation layers changes. When the number of feature transformation layers is one, it corresponds to the case that the total layer number is 10 and the decoupling rate is 90%. in Figure 3b. From Figure 3c, we can see that the optimal decoupling rate is different for different model depths, and that the differences in node classification accuracy under different decoupling rates can be as much as 40%. Therefore, although both JKnet and ResGCN have the ability to alleviate over-smoothing, their performances are still limited decoupling rates, and an approach to automatically control the decoupling rate is necessary.

## 4. Experiment

In this section, we perform experiments on node classifications to verify that our proposed framework can effectively overcome over-smoothing problems when deepening the backbone GNNs, which significantly improves the performance.

### 4.1. Dataset

In our experiments, assortative datasets and disassortative datasets are both verified. Neighboring nodes are more likely to be in the same category in assortative networks.

Citation networks such as Cora, Citeseer, and Pubmed are assortative benchmark datasets [37]. Each edge in these networks represents a citation relationship between two research papers, and node features are the bag-of-words vectors of corresponding paper abstracts. Each label indicates the category that the corresponding paper belongs to. Webpage networks such as Texas, Cornell, Wisconsin, and Chameleon are disassortative benchmark datasets. Edges in these networks represent hyperlinks between two web pages, and features of nodes contain webpage information. Each label indicates the category that the corresponding webpage belongs to.

Details of these datasets are recorded in Table 1.

### 4.2. Baseline and Setting

We compare our framework with a number of baseline approaches, including GCN, GAT [5,17,22,23,31], which are shallow models that are prone to have over-smoothing and over-fitting issues. JKnet, IncepGCN, and APPNP [17,22,31] are the models that try to alleviate over-smoothing by modify the propagation structure. DAGNN is a model that tries to overcome over-smoothing through decoupling feature transformation and propagation [30]. In order to verify that our framework can not only extend the depth of any kinds of GNNs, but also improve their performance to outperform existing deep models, we adopt a variety of backbones: GCN, JKnet, IncepGCN, and APPNP.

We use the Adam SGD optimizer with a learning rate of 0.01 and an early stopping patience of 100 epochs. We set the weight L2 regularization as 5 $\times 10^{-4}$, the dropout of shared MLP as 0.6, and the dropout of MLP corresponding to each block as 0.2.

### 4.3. Experimental Results Analysis

For the semi-supervised node classification task, we use Cora, Citeseer, and PubMed datasets, applying the standard fixed training/verification/testing splitting with 20 nodes per class for training, 500 nodes for validation, and 1000 nodes for testing [38]. For the fully supervised node classification, we use seven datasets: the Cora, Citeseer, PubMed, Chameleon, Texas, Cornell, and Wisconsin datasets. We randomly split nodes of each class into 60%, 20%, and 20% for training, validation, and testing, respectively. For each experiment, we run them 10 times and report the mean classification accuracy. Table 2 and Table 3 show the results.

Our proposed framework has superior generality. Whether the backbone GNN is likely to over-smooth, it can significantly improve its performance and outperform existing deep models. The results on both semi-supervised and fully supervised tasks confirm our view. It can effectively utilize deep model architectures to extract features from higher-order neighbors. This performance gain is due to the decoupling blocks that can aggregate multi-hop neighborhood features with adaptive depth, and the adaptive node receptive fields that allow the model to adaptively adjust the ratio of low-pass and high-pass node information.

### 4.4. Model Depth Analysis

In order to verify that our framework can alleviate over-smoothing issues with too many layers, we compare our GCN-BBAD with the Vanilla GCN and the DropEdge GCN under the same model depth. DropEdge is a framework for increasing model depth by randomly dropping edges [22]. Figure 4 reports the results of comparative experiments with different model depth. We perform experiments with 2/4/8/16/32/64 network layers on datasets, including Cora, Citeseer, and Pubmed. The performance of the Vanilla GCN degrades rapidly when the depth exceeds four layers. Although DropEdge performs better than Vanilla GCN, our framework can significantly reduce the over-smoothing issue. We attribute this phenomenon to the adaptive decoupling rate and the adaptive propagation depth in each block. As shown in Table 2, our framework is also applicable to complex backbone models such as JKnet, IncepGCN, and APPNP.

## 5. Conclusions

In this paper, we propose a novel Block-Based Adaptive Decoupling framework. Our framework utilizes adaptive decoupling blocks instead of multiple layers, which removes redundant feature transformation operations. We also propose a method based on identity mapping to automatically tune feature propagation depth within each block. We assign personalized node receptive fields to different nodes to effectively alleviate the over-smoothing issue. We theoretically identified that our blocks can provide diversified outputs, and we prove the effectiveness of the adoptive decoupling rate on over-smoothing. We demonstrate the importance of the decoupling rate. The experimental results verify our framework. This framework can also be used for many backbone networks to improve their performance.

## Figures and Tables

**Figure 1 entropy-24-01190-f001:**
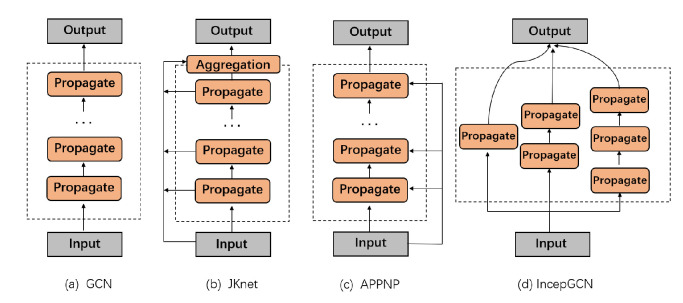
The illustration of four backbone networks. Benefiting from the flexibility and generality of our framework, each block in our framework can use a different backbone network.

**Figure 2 entropy-24-01190-f002:**
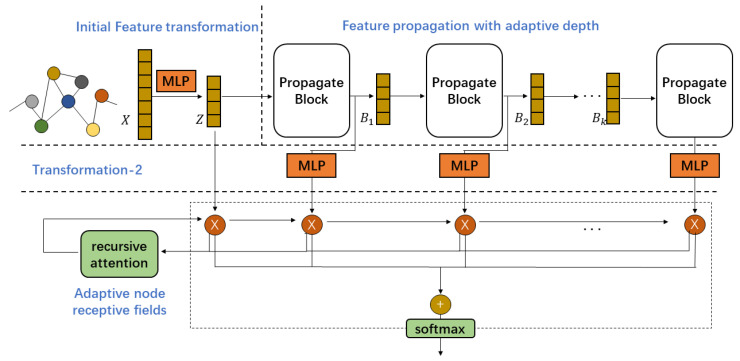
A visual illustration of our framework. It consists of feature transformations, feature propagations with adaptive depth, and flexible node receptive fields. Through block-based feature propagation with adaptive depth, we can adjust the decoupling rate automatically without redundant feature transformation layers. The receptive field of each node is adaptively adjusted using recurrent attention to obtain a personalized representation.

**Figure 3 entropy-24-01190-f003:**
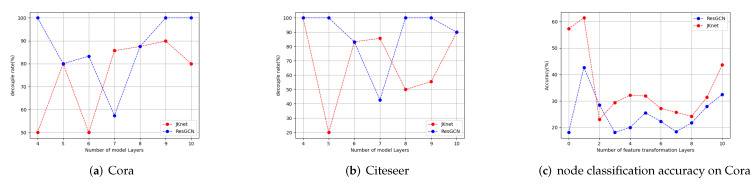
(**a**,**b**) Optimum decoupling rates with different depths in Cora and Citeseer. (**c**) Different feature transformation layers corresponding to the node classification accuracy on Cora.

**Figure 4 entropy-24-01190-f004:**
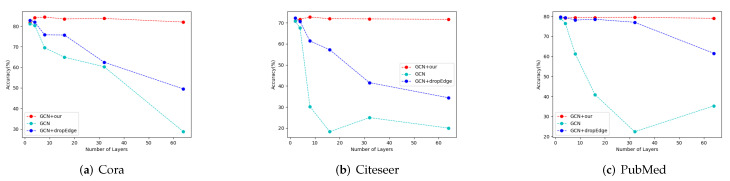
Accuracy comparison for three datasets with various depths.

**Table 1 entropy-24-01190-t001:** Datasets.

Dataset	Class	Nodes	Edges	Features
Cora	7	2708	5429	1433
Citeseer	6	3327	4732	3703
Pubmed	3	19,717	44,338	500
Chameleon	4	2277	36,101	2325
Cornell	5	183	295	1703
Texas	5	183	309	1703
Wisconsin	5	251	499	1703

**Table 2 entropy-24-01190-t002:** Comparison on semi-supervised node classification in terms of accuracy.

Method	Cora	Citeseer	Pubmed
GCN	81.5	71.1	79.0
GAT	83.1	70.8	78.5
APPNP	83.3	71.8	80.1
JKnet	81.1	69.8	78.1
DAGNN	84.4	73.3	80.5
IncepGCN	81.7	70.2	77.9
GCN-BBAD	84.3 (0.18)	72.6 (0.56)	79.9 (0.25)
APPNP-BBAD	**84.9** (0.10)	**73.4** (0.51)	**80.9** (0.14)
JKnet-BBAD	83.6 (0.48)	71.6 (0.11)	79.5 (0.04)
Incep-BBAD	83.3 (0.29)	72.0 (0.17)	79.5 (0.08)

**Table 3 entropy-24-01190-t003:** Comparison on fully supervised node classification in terms of AC.

Method	Cora	Citeseer	Pubmed	Texas	Cornell	Wisconsin	Chameleon
GCN	85.77	73.68	88.13	52.16	52.70	45.88	28.18
GAT	86.37	74.32	87.62	58.38	54.32	49.41	42.93
APPNP	87.87	76.53	89.40	65.41	73.51	69.02	54.30
JKnet	85.25	75.85	88.94	56.49	57.30	48.82	60.07
DAGNN	87.83	76.86	87.64	57.30	59.19	56.08	52.21
IncepGCN(Drop)	86.86	76.83	89.18	57.84	61.62	50.20	61.71
GCN-BBAD	87.42	75.91	88.25	72.73	72.97	77.45	54.91
APPNP-BBAD	**88.09**	**77.03**	89.63	**80.54**	75.95	81.57	59.12
JKnet-BBAD	87.38	74.62	88.99	79.73	81.08	84.31	60.81
Incep-BBAD	87.75	76.87	**89.70**	79.19	**81.62**	**84.51**	**62.24**

## Data Availability

Not report any data.

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
