# Peer review of "A Block-Based Adaptive Decoupling Framework for Graph Neural Networks"

_entropy, 2022, doi:10.3390/e24091190_

Round 1

Reviewer 1 Report

With the goal of reducing the issue of oversmoothing during the learning dynamics of Graph Neural Networks (GNNs), the authors propose a method to decouple the feature transformation and propagation processes. In detail, architectural blocks are introduced combining a single feature transformation and multiple propagation steps. Finally, an attention mechanism weights the blocks  contribution to the final node representation. 

The experimental section confirms the authors claims. Tha paper is well written , but the model description and notation lacks some details that hinder the overall proposal comprehension.

A) The paper could better describe, mostly in the related work section, the importance of the separation/decoupling of architectural depth (feature transformation) and diffusion (feature propagation) in related , such as  [1-3] and references therein. 

B) In Section 3.1.2 the authors use the notation m_v to denote the aggregated message from the neighboorhood. In the following lines (120) the authors use a different notation m_u.  Moreover, Eq. 9 is in matrix form while the previouses are in vectorial form.

C)  The exploited architectural block and its composition is not clearly defined in the paper.  How many feature transformations and propagations it is composed of? Also, the relation with the concept of layer/iteration in standard GNNs  is missing. This causes some issues in understanding the overall contribution. 

D)  Section 3.2 is not clear to me, in particular the concept of "different propagation structures". Was the DGCN model the authors refer to previousy introduced? 

E) In the experiments, the authors should report the results with an improved statistical assurance (more runs and standard deviation). 

[1] Zeng, Hanqing, et al. "Decoupling the depth and scope of graph neural networks." Advances in Neural Information Processing Systems 34 (2021): 19665-19679.

[2] Tiezzi, Matteo, et al. "Deep constraint-based propagation in graph neural networks." IEEE Transactions on Pattern Analysis and Machine Intelligence 44.2 (2021): 727-739.

[3] Dong, Hande, et al. "On the equivalence of decoupled graph convolution network and label propagation." Proceedings of the Web Conference 2021. 2021.

Reviewer 2 Report

I have marked up your paper and my annotations (in red) indicate a few places where I found potential (minor) errata.  Doubtless there are more and you should check carefully.  Your paper would be much improved (from the perspective of a reader, like myself, who is not knowledgeable about the various engineering aspects of neural networks which you cite) if you could define or sketch briefly in an appendix: What is a graph neural network, what is a block decomposition and a backbone network, what are feature transformation and propagation?  Ideally, give an interesting example and walk through how your methods apply.   Also, your formula for rate of decoupling, (45) on p 9, is rather strange.  If L_t = L_p, you get a rate of 0, while L_p < L_t would give a negative rate.  You should justify the formula more thoroughly.
